# Fabrication of Stretchable Transparent Electrode by Utilizing Self-Induced Vacuum Force

**Chunghwan Lee, Jaesool Shim, Chulho Bae \* and Kisoo Yoo \***

School of Mechanical Engineering, Yeungnam University, Gyeongsan 38541, Korea; grand1124@nate.com (C.L.);
jshim@ynu.ac.kr (J.S.)

**\*** Correspondence: chbai@yu.ac.kr (C.B.); kisooyoo@yu.ac.kr (K.Y.)

**Abstract:** The key challenge in fabricating a stretchable transparent electrode is the effective transfer of an electric conductor to a stretchable substrate. To this end, we used vacuum force to fully permeate the elastomer substrate into the electric conductor. The vacuum force was self-induced from the evaporation of the solvent in the electric conductor. Hence, a solvent, having a high evaporation rate, is postulated to exhibit superior fabrication quality. To demonstrate this, three different solvents were tested for preparation of the conductor slurry. In the test, the high-vapor-pressure solvents resulted in the superior quality of the fabricated stretchable electrode. Furthermore, the heating direction was changed during thermal curing to maximize the self-induced vacuum force. The plate-heating curing exhibited better transferring efficiency of the electric conductor because the evaporation of the solvent in the conductor slurry was accelerated faster than that of the thermal curing of the elastomer substrate. Besides the achieved high quality of the electrode, the fabrication cost can be drastically reduced because the extra process required to dry the electric conductor is omitted by simultaneous curing of the electric conductor and the stretchable elastomer substrate.

**Keywords:** stretchable transparent electrode; vacuum force; highly porous carbon

## 1. Introduction

Stretchable electrodes have received attention because of the increasing demands of flexible and stretchable devices, such as artificial skin and muscle, smart clothing, and electronic textiles [1,2]. Stretchable electrodes are fabricated by patterning an electric circuit on a stretchable substrate. Poly-di-methyl-siloxane (PDMS) has been widely used as an electrode substrate because of its stretchability and biocompatibility [3]. However, a main challenge for the fabrication process is in utilizing the flexibility of PDMS with electrically conductive materials, which are mostly inflexible. Therefore, numerous types of stretchable conductors have been studied. In addition, the production procedure of stretchable conductors can be categorized into two methods: The use of intrinsically stretchable conducting composites [4–6], and building the structure of the conductor such that bending and stretching are enabled [7]. Here, nanoparticle–polymer composites can be specified as intrinsically stretchable conducting composites. In nanoparticle–polymer composites, metal powder filler is widely used as the nanoparticles because of its simplicity of fabrication and high conductivity [8].

Another challenge in the fabrication of stretchable electrodes is patterning of the electric circuit, which has microscale dimensions. Owing to the high level of precision required, the stretchable conductor pattern has been prepared using the photolithography method [9]. Furthermore, other methods, such as growing the electrode pattern using chemical etching or by performing deposition with masking, have been proposed [10,11]. Despite achieving accurate circuit patterning, the abovementioned methods are too expensive for commercial use of fabricating stretchable electronic devices. Therefore, to overcome the cost issues, printed-electronics methods have been proposed

because of their simplicity of use in the circuit-patterning process and cost efficiency [12]. In printed electronics methods, the conductor materials are directly transferred or injected on an elastomer substrate. For example, the gravure-offset method has been highlighted because of its cost efficiency and rapid processing [13]. Nevertheless, the lithography method is still in development, as the adhesive force between the conductor to be transferred and the substrate is not sufficient, thereby, results in poor durability of the stretchable electrode.

In this study, we propose a fabrication method that, not only utilizes the flexibility of an electric conductor, but also improves the adhesive force between the electrode conductor and the elastomer substrate. In the proposed method, an electric conductor fills the engraved pattern on a stretchable substrate. To completely transfer the electric conductor material onto the substrate, we induced vacuum force, which thereby, resulted in full penetration of the elastomer into the electric conductor. Here, the vacuum force was self-induced, without the use of any vacuuming devices, because of the evaporation of the solvent in the conductor slurry. This self-induced vacuum force can only be achieved via the wet transferring process demonstrated in this study. An additional advantage of the proposed method is the significant reduction in the incurred cost of electrode fabrication, as the additional drying process for the electric conductor is omitted by transfer of the conductive slurry to the elastomer substrate under wet conditions (see Figure 1).

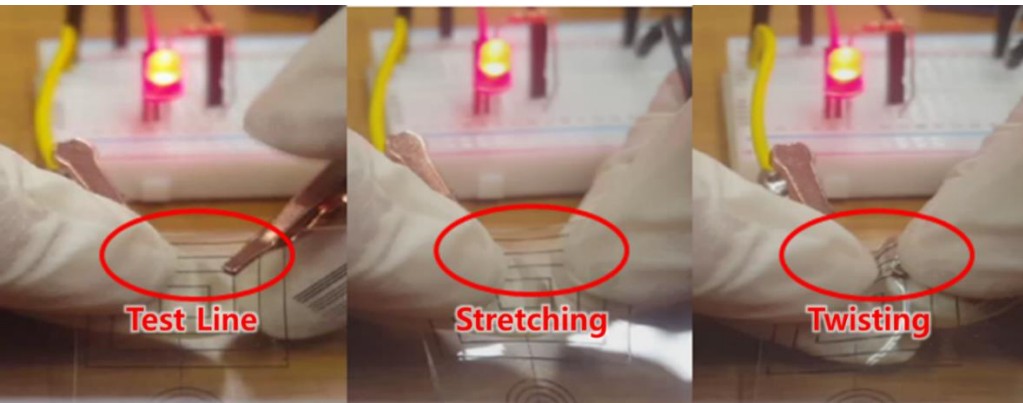

**Figure 1.** Mechanical test for the stretchable transparent electrode that is fabricated using the proposed method. The electrode exhibited stable electric conduction under stretching and twisting deformations.

## 2. Fabrication Methods

In this study, we used the lithography method to fabricate the stretchable electrode. An intaglio-patterned substrate was prepared using a silicon wafer, following which the patterns were filled with a conductive slurry. The conductor was then transferred to the elastomer substrate, and finally the stretchable electrode was fabricated.

### 2.1. Line-Patterned Substrate

To prepare the stretchable electrode, an intaglio-patterned substrate was developed using a silicon wafer. Here, the patterns were fabricated in rectangular shapes with different line widths, depths, and distances between the lines. The pattern was established using the conventional photolithography method. Figure 2 depicts the procedure for the preparation of the intaglio-patterned mold on the silicon wafer. First, a positive photoresist, a light-sensitive material, was deposited on the silicon wafer using spin-coating (3000 rpm). After coating, the photoresist was exposed to UV radiation via a mask; this mask was used to pattern the electrode lines. Subsequently, dry etching was performed to engrave the rectangular pattern on the silicon wafer. Finally, the mask was removed, and a mold, with an electrode pattern, was obtained. In this study, the intaglio-patterned mold was repeatedly used for the

fabrication of the stretchable electrode, which is one advantage of using the lithography technique for the fabrication.

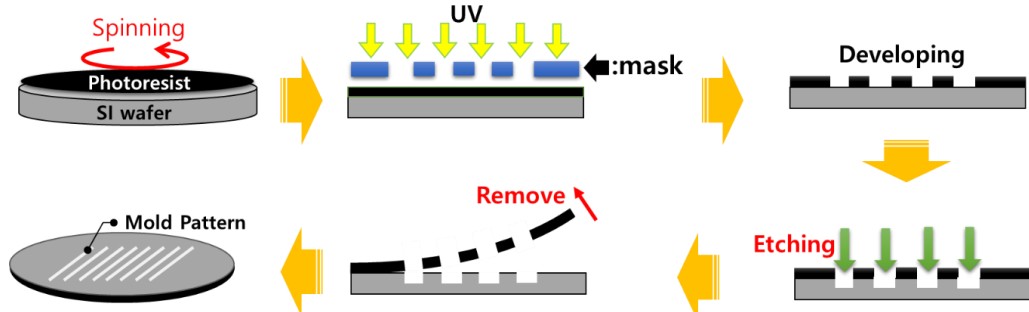

**Figure 2.** Preparation of intaglio-patterned mold on a silicon wafer. The procedure is based on the typical photolithography method [14].

### 2.2. Preparation of Composite Slurry and Elastomer

A porous-carbon particle and binder were mixed in a solvent for the preparation of the composite slurry to produce flexible and stretchable electrodes. First, two types of carbon nanoparticles, namely, a Denka Black granule (Denka Singapore Pte Ltd., Singapore) and Ketjen Black EC600 JD (Lion specialty chemicals Co., Ltd., Tokyo, Japan) were used for the preparation of the slurry. Ketjenblack EC is a carbon black that has excellent electrical properties compared with other carbon blacks, in general, due to its very good conductivity. This property has particular advantages in compounding to achieve conductivity. The superior property of Ketjenblack EC allows the preparation of relatively free compounding formulations to achieve the desired properties in this study. The Denka Black granule is known for its uniform particle size and good dispersion characteristic. Moreover, the Ketjen Black has very high absorption/desorption characteristics, as well as superior electric conductivity (Ketjen Black: 3.9 $\Omega$cm/conventional Acetylene Black: $1 \times 10^8$ $\Omega$cm). The performances of the stretchable electrodes, fabricated using each of the two types of carbon nanoparticles, will be discussed in Section 3.1. Polyvinylidene fluoride (PVdF), which is highly non-reactive thermoplastic fluoropolymer, is used as the binder. It was purchased from Sigma Aldrich. Subsequently, the carbon nanoparticles were dispersed into the solvent to establish a homogenous linkage between the binder and the carbon nanoparticles. Here, various solvents, such as acetone, tetrahydrofuran (THF), and ethanol, were used; the effect of each of these solvents will be discussed in Section 3.2. To ensure homogenous mixing and dispersion, the slurry was sonicated for 5 min. In this study, PDMS was used as the elastomer to fabricate the stretchable electrodes. Sylgard 184, which is a silicon elastomer kit, was purchased from Dow Corning, and the mixing ratio of the base to the curing agent was set to 10:1 for gelation and conversion to a solid elastomer.

### 2.3. Fabrication Procedure

A stretchable electrode was fabricated by transferring the electrically conductive line from the silicon wafer to the elastomer. Figure 3 depicts the fabrication procedure in detail. First, the prepared composite slurry was poured onto the silicon wafer, on which the electrode lines were etched in advance (step 1). Doctor blades were used on the silicon wafer to fill the slurry into the intaglio line pattern, as well as remove the excess slurry from the surface of the silicon wafer (step 2). After this, the silicon wafer was immediately coated with the thermoset elastomer PDMS, while the slurry was still wet (step 3). The elastomer was then cured in a convection oven at 100 °C for 10 min (step 4). Following the curing, the elastomer was peeled off, and the electrode lines were transferred from the silicon wafer to the elastomer (step 5). Finally, the fabrication procedure of the stretchable electrode was completed. There is a broad distinction between the proposed method and the conventional

electrode-fabrication method. In the conventional method, the elastomer is applied onto the solid electrode pattern, following which the elastomer is cured using UV exposition or thermosetting. Meanwhile, in the proposed method, the thermoset elastomer is coated on the wet electrode pattern, then, the elastomer and the electrode are heated concurrently. In this way, a vacuum force is developed in the intaglio pattern; therefore, full penetration of the elastomer into the pattern is achieved.

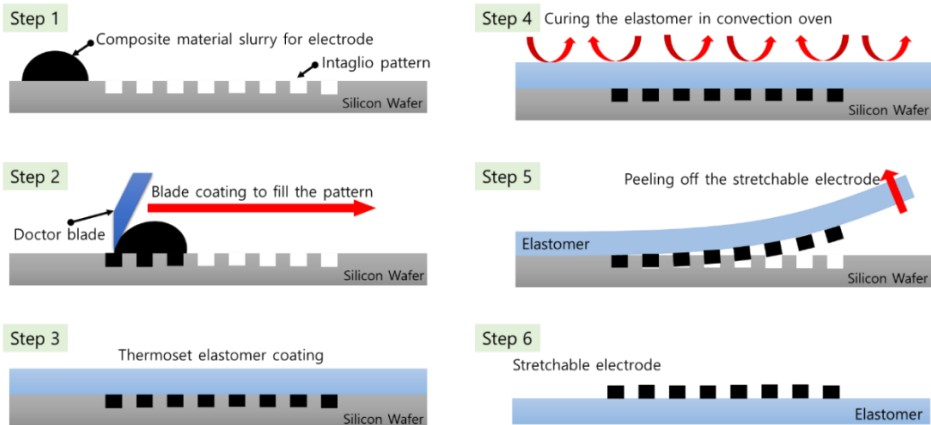

**Figure 3.** Fabrication procedure of a stretchable electrode. Intaglio pattern is filled with electrically conductive material and then transferred to an elastomer substrate because of the high adhesive force between the electrode conductor and the elastomer substrate, in the peeling-off step. The self-induced vacuum force resulting in the penetration of the elastomer into the pattern is driven during the curing process (step 4).

Figure 4 depicts the phenomena during the heating process. Initially, the pore area is filled with volatile solvent. During the heating process, the evaporation of the solvent is accelerated, thereby, inducing vacuum in the pores. Because of this self-induced vacuum in the pores, both the full penetration of the highly viscous elastomer into the pattern and the excellent transfer ratio of the electrode pattern, from the patterned silicon substrate to the elastomer, are achieved.

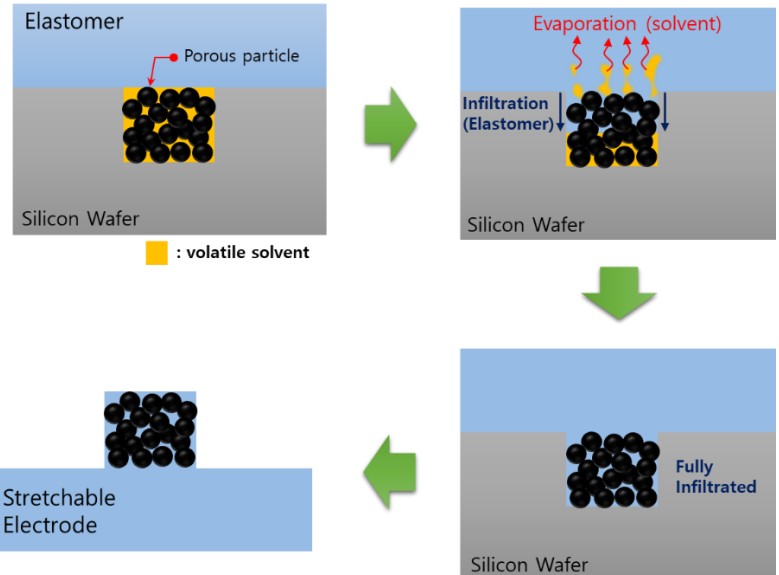

**Figure 4.** Fabrication step 4. The self-induced vacuum force is driven by the available pores resulting from the evaporation of the solvent. Subsequently, the pore space is filled with the elastomer until the substrate is fully cured.

## 3. Results and Discussion

### 3.1. Feasibility of the Proposed Fabrication Method

The stretchable electrode was fabricated using the methods described in the previous section. The Ketjen Black and PDMS were used to prepare the electrically conductive particle, and the elastomer, respectively. Here, 50-, 100-, and 150-μm wide electrodes were fabricated. As Figure 5a depicts, the stretchable electrode was well fabricated using a highly viscous elastomer (e.g., PDMS) on an intaglio-patterned substrate. The elastomer was able to fully penetrate the line pattern, thereby, covering the electrically conductive particles. Note that Figure 5b, obtained using the optical three-dimensional (3D) profiler, indicates that carbon black (Ketjen Black) exists only at the intaglio-patterned area (red line). Furthermore, it was observed that the full penetration of the elastomer into the line pattern resulted in a high transfer ratio of the electrode pattern from the pattern substrate to the elastomer. Additionally, note that Figure 5c, obtained using confocal laser scanning, indicates that the line profile is straight at the top of the electrode.

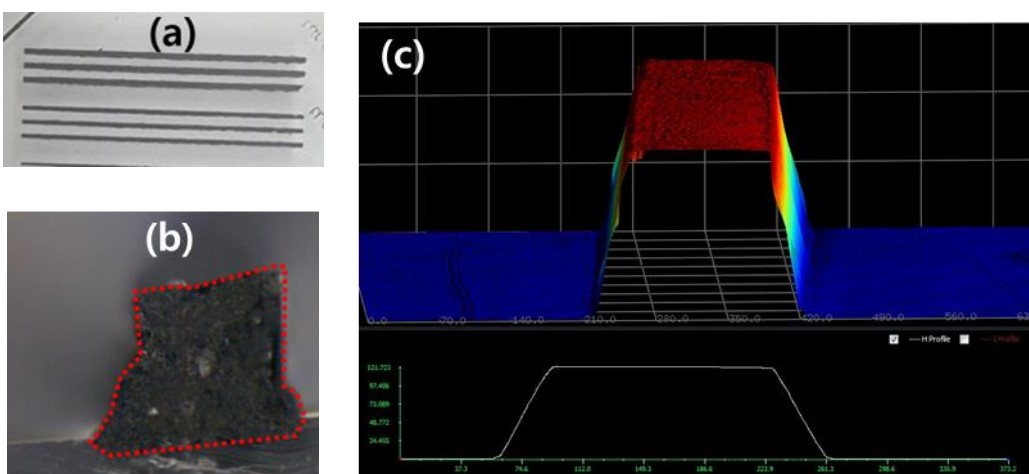

**Figure 5.** (**a**) Electric conductor pattern on an elastomer (top-view) (**b**) Section image of the conductive pattern on the stretchable elastomer substrate. (**c**) 3D optical profiler image of the pattern. The thickness and the aspect ratio (line depth to width) of the pattern is measured using a three-dimensional (3D) optical profiler.

The cross-sectional, scanning-electron-microscopy (SEM) images of the 50-, 100-, and 150-μm wide electrode lines were investigated. Here, the images and EDS data obtained from S-4800; Hitach Instrument. As Figure 6a–c depict, clear section profiles were obtained irrespective of the electrode-line width. It is noteworthy that a high electrode-transfer ratio could be achieved even for a high aspect ratio (line depth to width), which is an unsuitable condition for the penetration of the elastomer into the electrode pattern. The SEM-energy-dispersive X-ray spectroscopy (EDX) test shows the elastomer penetration. High carbon contents were observed at levels 2, 3, and 4, where the carbon-black particles were expected to exist, while the EDX analysis indicates lower carbon contents at levels 1 and 5, where the elastomer enclosed the particles (see Figure 6d). Obviously, a higher content of carbon particles was observed in the core of electric conductor (Levels. 2–4), while a lower value at the edge of the electric conductor (Levels. 1 and 5) was observed, due to the dispersion of the carbon materials. The difference of carbon content is in 10% roughly.

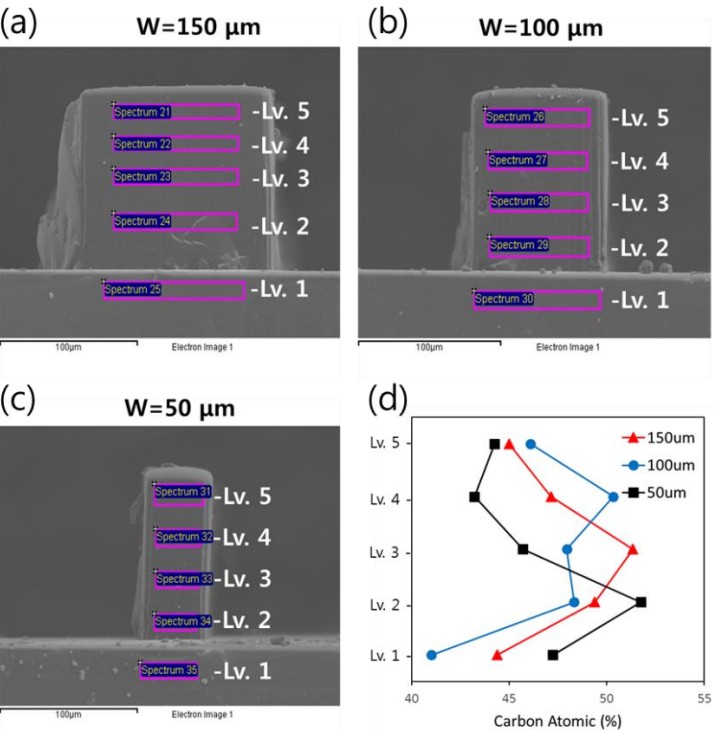

**Figure 6.** Scanning-electron-microscopy (SEM) image of the cross-section of the electric conductor for (**a**) 150 μm (**b**) 100 μm (**c**) 50 μm width. (**d**) carbon content of Levels. 1–5. The ratio of carbon at different positions is obtained from the SEM–EDX analysis.

Finally, the mechanical test for the stretchable electrode was performed by stretching and twisting the electrode, while conducting electricity through it (recall Figure 1). From the test, it is concluded that the proposed method enables the fabrication of stretchable electrodes with a highly viscous electrode.

### 3.2. Advantage of Vacuum Force on Elastomer Penetration

In this section, the effect of the self-induced vacuum force on the fabrication of the stretchable electrode will be discussed. As previously mentioned, the self-induced vacuum force expedites the penetration of the elastomer into the electrode pattern, such that the electrode transfer becomes feasible even for highly viscous elastomers. Because the vacuum force is induced by the evaporation of the solvent, the solvent should be sufficiently absorbed by the electrode particles. Here, two types of carbon nanoparticles (Ketjen Black and Denka Black) were used for the electrode fabrication. Ketjen Black is a well-known carbon-based material because of its highly porous structure and electric conductivity. To show the adsorption/desorption performances of the solvent, BET(Brunauer-Emmett-Teller)/BJT(Barrett-Joyner-Halenda) tests were conducted for both these carbon materials. From Table 1, it can be observed that Ketjen Black exhibited 10 times superior adsorption/desorption characteristics than those of Denka Black. Hence, it was decided that the proposed fabrication method, which uses the self-induced vacuum force, would be performed using the electrode slurry prepared from the Ketjen Black nanoparticles.

**Table 1.** BET(Brunauer-Emmett-Teller)/BJT(Barrett-Joyner-Halenda) test results for the porous electrically conductive materials (Denka Black/Ketjen Black).

| Test | Denka Black | Ketjen Black |
|---|---|---|
| BET surface area (m$^2$/g) | 67.9778 | 894.0720 |
| BJH adsorption surface area of pores (cm$^2$/g) | 0.1788 | 1.0830 |
| BJH desorption surface area of pores (cm$^2$/g) | 0.1781 | 1.0994 |

Figure 7a,b depict the SEM images for sections of the electrode lines fabricated using Ketjen Black and Denka Black, respectively. The full penetration of the elastomer occurs for the Ketjen Black slurry (note the red-dotted line in Figure 7a). However, the elastomer could not fully penetrate the electrode of the Denka-Black-based slurry at the pattern corners. From this comparison, it can be concluded that the electrode slurry should be prepared using highly porous carbon nanoparticles along with a sufficient amount of a volatile solvent that induces a vacuum force, which, in turn, enables the full penetration of the elastomer into the pattern.

(**a**) Ketjen Black                                (**b**) Denka Black

**Figure 7.** Cross-section SEM image for electric conductor pattern prepared in (**a**) Ketjen Black and (**b**) Denka Black showing the characteristic of elastomer penetration into porous conductive materials on varying the porosity. Compared with Denka Black, the Ketjen Black has higher adsorption ability, thereby leading to superior penetration of the elastomer in the case of using the latter.

### 3.3. Effect of Evaporation Rate on the Fabrication Process

As discussed in the previous sections, a volatile solvent adsorbed onto the carbon particles is critical to inducing a vacuum force, and results in the penetration of the highly viscous elastomer into the intaglio pattern. However, the entire solvent should be evaporated during the thermosetting step before the elastomer is cured. If the vaporized solvent does not fully escape until the elastomer is cured, gases will be trapped in the cured elastomer, causing the bubbling defect. Furthermore, the vaporized solvent would be trapped in the elastomer if the viscosity of the elastomer is too high, even before the elastomer is cured. Because the viscosity of a thermoset elastomer increases upon heating, it is crucial that the solvent should be fully evaporated before the viscosity reaches the critical value. Hence, from Figure 8, it can be proposed that a stretchable electrode without the bubbling defect will be obtained under the condition of Case-A, while the same defect will be observed for the condition of Case-B.

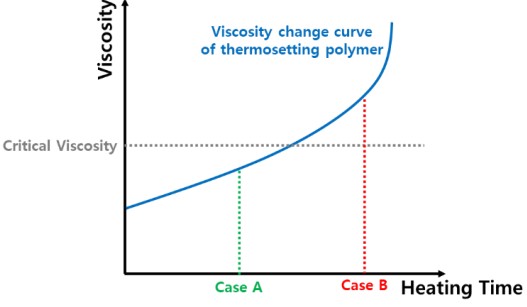

**Figure 8.** Schematic viscosity–temperature curve of the elastomer during thermal curing. Full evaporation of the solvent in the porous conductive material is desired before reaching the critical viscosity of the elastomer (Case A) to avoid the trapping of the vaporized solvent into the elastomer (Case B).

For demonstration, several solvents with different evaporation rates were tested by performing the proposed fabrication method. Three solvents, namely, acetone, THF, and ethanol, were used to prepare carbon slurry. Among the solvents, acetone exhibited the highest evaporation rate of 7.7, where the evaporation rate of butyl acetate is defined as 1.0. Meanwhile, ethanol exhibited the lowest evaporation rate of 1.7. Furthermore, the vapor pressure of acetone was the highest at 24 kPa, while the vapor pressure of ethanol was lower than those of acetone and THF. The detailed properties related to evaporation are presented in Table 2. Here, it was anticipated that the carbon slurry, based on ethanol, would cause the bubble defect in the cured elastomer due to the inferior characteristic of evaporation for ethanol.

**Table 2.** Vaporization ability for different solvents used to prepare electrically conductive slurry [15].

| Solvent | Evaporation Rate (n-Butyl Acetate = 1.0) | Boiling Point | Vapor Pressure (20 °C) |
|---------|------------------------------------------|---------------|------------------------|
| acetone | 7.7 | 55.1 °C | 24.0 kPa |
| THF | 8.0 | 66.0 °C | 143 kPa |
| ethanol | 2.3 | 78.2 °C | 5.87 kPa |

Figure 9 depicts the stretchable electrodes fabricated using carbon slurry based on the use of the three different solvents separately. As expected, the bubble defect was observed in the electrode that used the carbon slurry based on the ethanol solvent; this defect resulted in poor electric conduction. According to the hypothesis, this fabrication condition is considered Case-B (note Figure 8). However, referring to the results, acetone and THF are feasible solvents for the proposed electrode-fabrication method (Case-A).

Acetone               THF               Ethanol

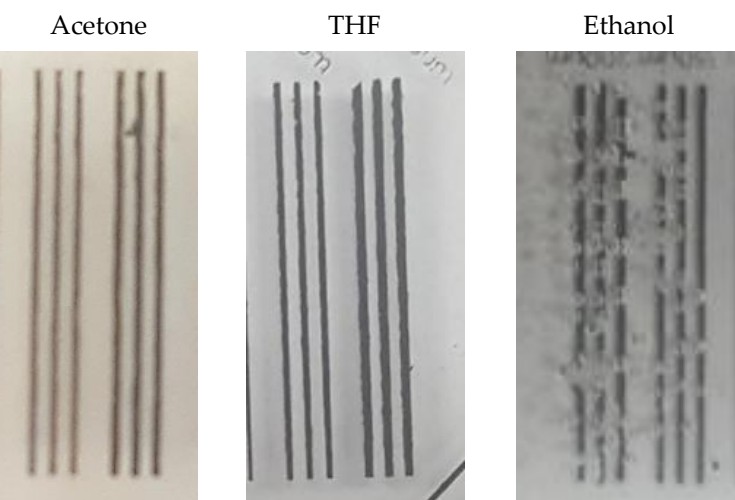

**Figure 9.** Fabrication quality of the stretchable electrodes fabricated using different slurry solvents. The void in the elastomer substrate is observed only in the case of using ethanol solvent, which has relatively poor vaporization ability (Case B in Figure 8).

Considering the characteristic of evaporation, acetone is a better solvent than THF. However, both the solvents show feasibility of the fabrication. Hence, we measured the electrical resistance of the electrodes, which also represents their electrical conductivities. To ensure reliability of the experiment, the average values of the resistances of 15 electrode samples were compared. In addition, the volume of the solvents in the synthesis of carbon slurry was varied from 12 to 17 mL in increments of 1 mL. Figure 10 depicts the averaged values of the resistances of the 15 samples with varying solvent volumes. The resistances were measured by simple two-probe DC technique (Fluke 15b Digital Multimeter). Both ends of the conductor were selected as measuring points. Here, the measurement were carried out under room temperature (25 °C). As expected from the table for the rates of evaporation, a superior

performance of electric conduction (i.e., lower resistance) was observed in the electrode using acetone as the solvent in the carbon slurry (note Figure 10a). From this comparison, the effect of the evaporation rate of the solvent on the fabrication process can be established.

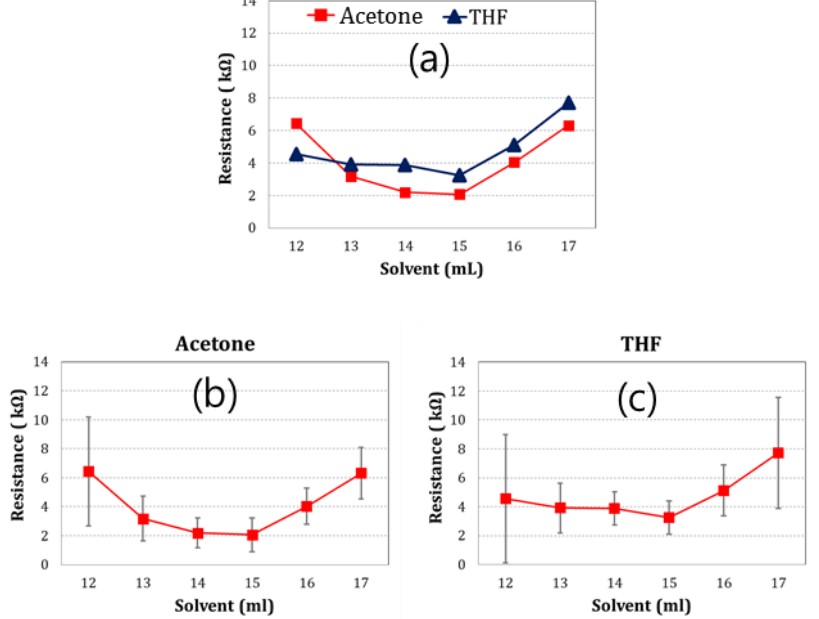

**Figure 10.** Electrical resistance of the electrode pattern depending upon the fraction of the solvent (**a**) Acetone and THF (**b**) Acetone (**c**) THF in the conductor slurry.

Additionally, for both acetone and THF, the lowest value of electrical resistance (i.e., highest value of electric conductivity) was obtained when the amount of solvent used to prepare the carbon slurry was 15 mL. However, the electrical resistance increased as the amount of solvents increased (note Figure 10a,b for 15, 16, and 17 mL of both the solvents). This increase in the electrical resistance was caused because the portion of the carbon particles, the main electric conductors in the electrode, decreased as the amount of solvent was increased. Consequently, the carbon particles became excessively isolated from the elastomer, thereby deteriorating the electrical-conductivity performance of the electrode. However, the resistance slightly increased as the portion of solvent decreased in the slurry from 15 to 13 mL. This increase in the resistance was observed because the self-induced vacuum force that causes the penetration of the elastomer into the line pattern was weakened as the amount of solvent decreased. This weakening of the vacuum force, in turn, reduced the transfer ratio of the electrode pattern from the pattern substrate to the elastomer, thereby resulting in lower electric conduction. These results suggest that there is an optimum proportion value of the solvent for the preparation of the carbon slurry. This value strikes a balance between the penetration of the elastomer and the isolation of carbon particles from the elastomer.

### 3.4. Effect of Heating Methods on the Fabrication Process

As previously mentioned, the evaporation of the solvent in the slurry should be completed before the viscosity of the elastomer reaches the critical value. Hence, it is preferable to heat the wet electrode prior to thermosetting the elastomer during the curing process. For this reason, the curing process was performed under various heating conditions: either in a convection oven or on a hot plate. In the former condition, the thermal energy was delivered by convection from hot air both to the elastomer and the silicon wafer. While, in the latter condition, the silicon substrate/wafer was heated first. Therefore, the evaporation of the solvent occurred earlier than the curing (see Figure 11). Hence, the penetration of the elastomer into the electrode was observed to accelerate when the silicon substrate was cured on the

hot plate. Owing to this, it was considered that using the hot-plate curing, electrode fabrication could be achieved, even with low-porous carbon or excessive solvent in the slurry.

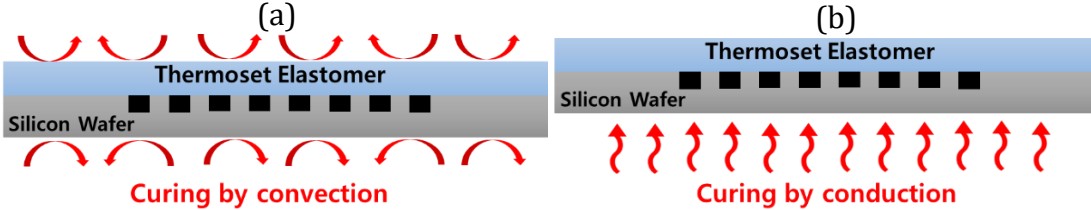

**Figure 11.** Thermal curing process using (**a**) convection heating from all directions and (**b**) conduction heating to the bottom.

To demonstrate this hypothesis, the electrodes were fabricated using the slurry synthesized with Denka Black. In Section 3.2, it was confirmed that only the vacuum force is insufficient to enable the full penetration of elastomer into the electrode slurry because of the poor absorption/desorption characteristics of Denka Black (recall Figure 7). Consequently, an asymmetric electrode was obtained using the Denka-Black-based slurry. However, a well-formed electrode was obtained by curing on a hot plate (see Figure 12). In Figure 12b, the uniformly distributed particles in the electrode area and along the straight boundary between the electrode and elastomer are confirmed. In the case of curing on a hot plate, the good fabrication quality can be attributed to the fact that heat transfer began from the bottom of the silicon substrate on the hot plate, following this, the solvent evaporation, which occurs prior to the curing of elastomer, resulted in the full penetration of the elastomer.

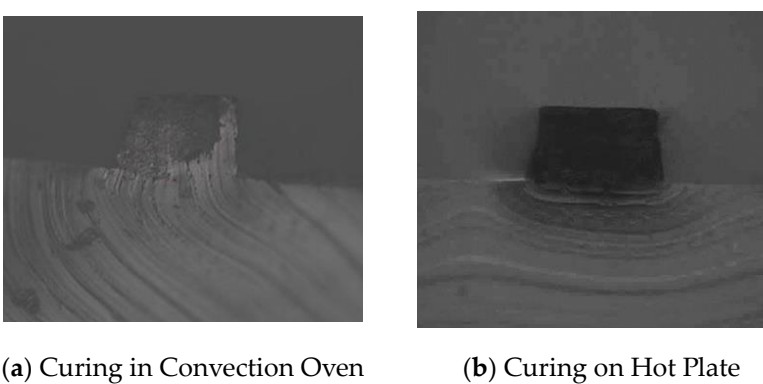

　　　(**a**) Curing in Convection Oven　　　　　　　(**b**) Curing on Hot Plate

**Figure 12.** SEM images to show the characteristic of elastomer penetration into porous conductive materials with different thermal curing methods: curing (**a**) in convection oven and (**b**) on hot plate. The heating of the bottom of the silicon substrate results in the solvent evaporation preceding the elastomer curing, thereby leading to superior penetration of the elastomer.

To demonstrate the effect of the heating method used, the fabrication process was also examined by changing the amount of the solvent in the electrode slurry. In this study, the stretchable electrodes were fabricated using slurries synthesized with 10, 13, and 23 mL acetone. First, the stretchable electrode was cured in a vacuum oven for convective heat transfer. As discussed in Section 3.3, a well-fabricated electrode was obtained using the slurry synthesized in 10- and 13-mL acetone due to its high volatility (see Figure 13a,b). However, the bubble defect was observed in the fabrication using 23 mL of acetone; note the bubbles in the elastomer shown in Figure 13c. Although, the volatility of acetone is superior, its excessive amount in the slurry hinders its complete evaporation (changing case A to case B in Figure 8). Therefore, the remaining acetone (i.e., the unevaporated acetone) in the electrode slurry was captured with the highly viscous elastomer during the curing process. Here, the electrode fabricated using the slurry synthesized with 23 mL of acetone was cured on a hot plate (100 °C). As Figure 14 depicts, the bubble defect that disappeared in the stretchable electrode, cured

on a hot plate, and the resistance of the electric conductor was then measured as 1 kohm (comparing Figure 14a,b). Hence, it can be concluded that the curing on a hot plate results in the evaporation of the solvent ahead of the curing of the thermoset elastomer. These results strengthen the elucidation of vacuum-force-driven elastomer penetration.

Thick Slurry　　　　　　　　　　　　　　　　　Thin Slurry
(**a**) Acetone (10 mL)　　　　(**b**) Acetone (13 mL)　　　　(**c**) Acetone (23 mL)

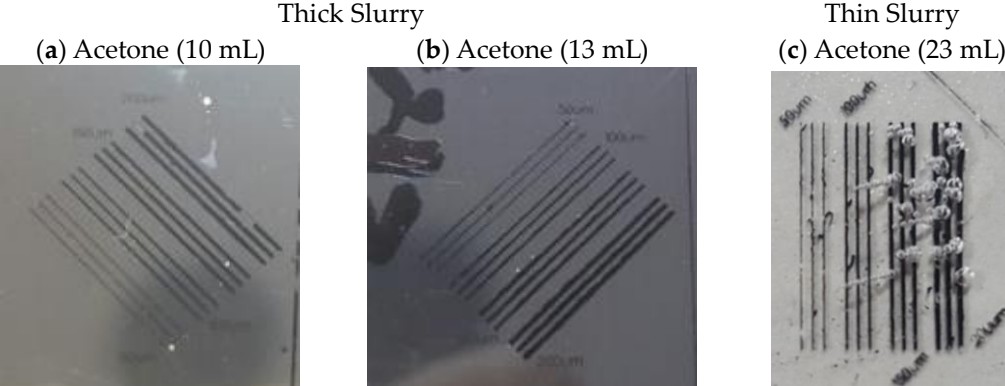

**Figure 13.** Solvent gas trapped by highly viscous elastomer during the thermal curing process in a convection oven for the slurry synthesized with excessive solvent. The void in the elastomer substrate is observed only for the fabrication with excessive amount of solvent in slurry (thin-slurry case).

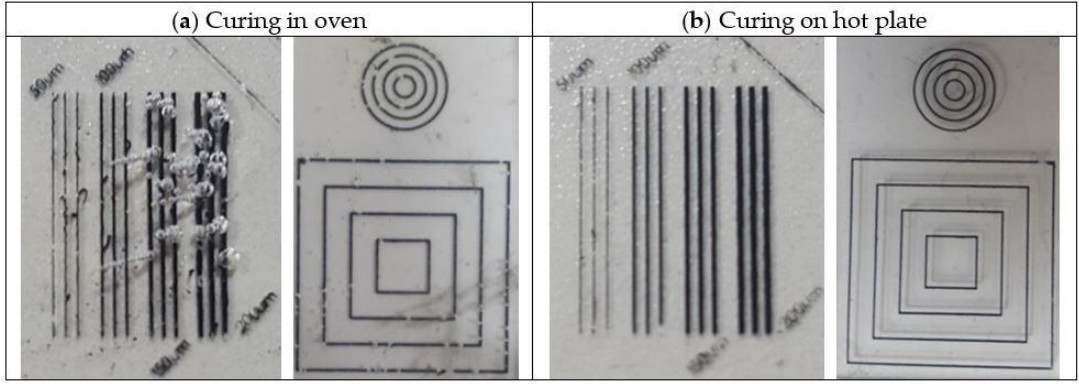

**Figure 14.** Advantage of the selective thermal curing on a hot plate. The void caused by the trapped solvent gas is disappeared after heating the conductive-slurry region prior to elastomer substrate.

## 4. Summary and Conclusions

A curing method, using a self-induced vacuum force for the fabrication of stretchable electrodes, was proposed in this study. The self-induced vacuum force inside the electrode conductor resulted in the full penetration of the elastomer into the electrode pattern. In the tests, the complete penetration of the elastomer into the intaglio pattern was only observed for the electrode fabricated using Ketjen Black, which has superior adsorption/desorption characteristics. Furthermore, the fabrication exhibited an extraordinary transfer ratio of the electrode conductor from the silicon substrate to the elastomer. However, in the case of the electrode slurry, synthesized using Denka Black, which has low adsorption/desorption characteristics, the elastomer could not penetrate the particles completely. This result indicates that the particles contained insufficient amounts of solvent due to their poor adsorption/desorption characteristics, thereby, resulting in a weak vacuum force.

Additionally, the effect of solvent volatility on the proposed method was investigated. Here, three solvents, namely, acetone, THF, and ethanol, were considered. When ethanol was used, which has inferior volatility compared with the others, bubble defects were observed in the cured elastomer. This is because ethanol could not evaporate completely before the elastomer was cured due to its inferior volatility. Therefore, the gasified solvent (ethanol) was captured in the elastomer. However, a clear

transparent electrode was fabricated using acetone and THF as the solvents, because of their higher evaporation rates than that of ethanol.

Finally, we analyzed the effect of the curing method used on the fabrication quality. The curing process was performed on a hot plate, and the resultant electrode was compared with an electrode cured in a convection oven. The full penetration of the elastomer into the particles was observed by curing the electrode on a hot plate, even with the inferior slurry using Denka Black. However, this level of fabrication quality could not be obtained by curing the elastomer in a convection oven. This result indicates that curing on a hot plate provokes the evaporation of the solvent prior to the thermosetting of the elastomer, thereby, further inducing the penetration of the elastomer. Based on these results, it can be concluded that the vacuum force is a key factor for achieving a high transfer ratio of the electrode conductor from the silicon substrate to the elastomer. Furthermore, the penetration can be enhanced by considering the adsorption/desorption characteristics of the particle, volatility of solvent, and curing method.

**Author Contributions:** Investigation, C.L., Writing-review & editing, J.S., Funding acquisition, C.B., Writing-original draft preparation, K.Y. This paper was prepared in the contributions of all authors.

**Funding:** This research received no external funding.

**Acknowledgments:** This work was supported by a 2016 Yeungnam University Research Grant.

**Conflicts of Interest:** The authors declare no conflict of interest.

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
