# Peer review of "Fabrication of Stretchable Transparent Electrode by Utilizing Self-Induced Vacuum Force"

_applsci, doi:10.3390/app9234986_

Round 1

Reviewer 1 Report

Please see the attached file of review comments

Author Response

Thanks for your valuable review. I thank the reviewer for their detailed and thoughtful comments and believe that my inclusion of their suggested revisions has strengthened the manuscript. I am very grateful for the assistance and consideration.

Point-by-Point Response are included in attachment.

I hope that it will now be acceptable for publication in this journal.

Reviewer 2 Report

 I do not understand the motivation behind this work. Getting a conducting line transferred from silicon grooves onto an elastomer is not new. Why are the resistances so high? What is the application of these lines if their resistances are so high? There is no scale bar in many of the images, which makes it difficult to look at the scale of the work. Some images are redundant for example Figure 10 has three images, which are not labelled and also the first figure is a combination of the other two. Then why to show three images when one is already showing the other two. There is error bar shown in two graphs and the first graph does not show any error bars.

This work needs a lot of work before it is considered for publication as it lacks the motivation with which this work is being made and the scientific basis and representation of the data.

Author Response

Thanks for your valuable review. I thank the reviewer for their detailed and thoughtful comments and believe that my inclusion of their suggested revisions has strengthened the manuscript. I am very grateful for the assistance and consideration.

Point-by-Point Responses are included in attachment.

I hope that it will now be acceptable for publication in this journal.

Round 2

Reviewer 1 Report

The authors have revise their paper according to the referee's comments.

Therefore, the paper could be accepted in the present form.